# Genetic Structuring of One of the Main Vectors of Sylvatic Yellow Fever: *Haemagogus* (*Conopostegus*) *leucocelaenus* (Diptera: Culicidae)

**DOI:** 10.3390/genes14091671

**Published:** 2023-08-24

**Authors:** Ramon Wilk-da-Silva, Antônio Ralph Medeiros-Sousa, Luis Filipe Mucci, Diego Peres Alonso, Marcus Vinicius Niz Alvarez, Paulo Eduardo Martins Ribolla, Mauro Toledo Marrelli

**Affiliations:** 1Institute of Tropical Medicine, University of São Paulo, São Paulo 05403-000, Brazil; 2Department of Epidemiology, School of Public Health, University of São Paulo, São Paulo 01246-904, Brazil; aralphms@yahoo.com.br (A.R.M.-S.); diego.p.alonso@unesp.br (D.P.A.); 3State Department of Health, Pasteur Institute, São Paulo 01027-000, Brazil; lfmucci@gmail.com; 4UNESP—Biotechnology Institute and Biosciences Institute, Sao Paulo State University, Botucatu 18618-689, Brazil; marcus.alvarez@unesp.br (M.V.N.A.); p.ribolla@unesp.br (P.E.M.R.)

**Keywords:** mitochondrial, *Haemagogus*, SNPs, Atlantic Forest, genetic diversity, sylvatic vector, yellow fever

## Abstract

Genetic diversity and population structuring for the species *Haemogogus leucocelaenus*, a sylvatic vector of yellow fever virus, were found to vary with the degree of agricultural land use and isolation of fragments of Atlantic Forest in municipalities in the state of São Paulo where specimens were collected. Genotyping of 115 mitochondrial SNPs showed that the populations with the highest indices of genetic diversity (polymorphic loci and mean pairwise differences between the sequences) are found in areas with high levels of agricultural land use (northeast of the State). Most populations exhibited statistically significant negative values for the Tajima D and Fu FS neutrality tests, suggesting recent expansion. The results show an association between genetic diversity in this species and the degree of agricultural land use in the sampled sites, as well as signs of population expansion of this species in most areas, particularly those with the highest forest edge densities. A clear association between population structuring and the distance between the sampled fragments (isolation by distance) was observed: samples from a large fragment of Atlantic Forest extending along the coast of the state of São Paulo exhibited greater similarity with each other than with populations in the northwest of the state.

## 1. Introduction

Tropical forests are home to most of the planet’s biodiversity [1]. Two examples are the Amazon Forest, which alone accounts for around one-quarter of all the species in the world [2], and the Atlantic Forest, one of the world’s biodiversity hotspots, where a very high concentration of endemic species are undergoing intense habitat loss [3]. These areas are also responsible for providing what are known as ecosystem services, a wide range of benefits that mankind obtains from nature, which are fundamental for its well-being [4], such as climate regulation, the provision of water, pollination and disease regulation [5].

The intense deforestation observed in tropical forests not only constitutes one of the main threats to species in these areas [6], but also jeopardizes the survival of the human species itself. This scenario has been reflected in the growing number of epidemics of zoonotic origin over the last decades, i.e., epidemics transmitted from animals to humans as a result of humans coming into contact with animals infected with pathogens [7,8].

In the particular case of zoonoses caused by pathogens transmitted by mosquitoes, spillover events are largely determined by the overlapping of hosts, vectors and human populations [9]. Sylvatic yellow fever cycle is an excellent model to study, since it involves several species of non-human primates (NHPs) and mosquitoes belonging to the genera *Haemagogus* and *Sabethes*. Anthropogenic landscape changes, such as deforestation and changes in land use, are the main reasons for the increase in the incidence of zoonoses, including sylvatic YF [10,11,12]. Indeed, around 40% of recent zoonotic diseases were the result of deforestation and the associated changes in land use [13].

Yellow fever (YF) is an infectious disease [14] caused by a virus of the genus Flavivirus (family Flaviviridae), which circulates in the wild among NHPs by means of mosquito vectors, particularly species in the genera *Haemagogus* and *Sabethes* in Central and South America and *Aedes* (subgenus *Diceromyia* and *Stegomyia*) on the African continent [14,15]. The disease has two different transmission cycles, one urban and the other sylvatic. The former involves interhuman transmission of yellow fever virus (YFV) by *Aedes aegypti*, while in the latter, humans are infected sporadically when they enter forested areas where the virus circulates between NHPs and wild mosquitoes [14]. An intermediate cycle (“zone of emergence”) has also been described, in which vectors can transmit the virus to both men and NHPs. These zones have been observed primarily in rural areas of Africa [16].

The species considered the main vectors in Brazil are *Haemagogus janthinomys* (Dyar 1921) and *Haemagogus leucocelaenus* (Dyar and Shannon, 1924) [17,18], both of which have a widespread distribution throughout Brazil, except in the state of Rio Grande do Sul, where to date, the former has not been recorded [19]. As for the eco-epidemiology of these species, *Hg. janthinomys* is frequently found infected with YFV in outbreaks of YF in the Southeast, Midwest and North (in the last of which it is endemic), whereas *Hg. leucocelaenus* is found in the Southeast and South [18,20,21,22,23].

An outbreak of sylvatic YF of unprecedented magnitude occurred in Brazil recently, and 2276 human cases and 2978 laboratory-confirmed epizootics were reported by the end of 2021 [24,25,26]. Of particular note among the factors that led to this epidemic are man-made landscape changes, such as changes in land use and deforestation, which are some of the primary causes of the increased incidence of pathogens of zoonotic origin [10,11,12].

In the particular case of deforestation and the consequent habitat fragmentation, the result is not only an increase in edge surface [27], which in turn implies overlapping of mosquitoes of the species *Hg. leucocelaenus* [28] and howler monkeys (*Alouatta* spp.) [29], but also a greater prevalence of YFV among these monkeys because of the greater population density of NHPs [30]. At the same time, increased forest edge surface can represent a larger area in which humans can come into contact with the virus itself [31]. Hence, fragmentation can be a key element in epidemic and epizootic dispersal of YFV, with forest edges playing an important role in the circulation of the virus [32,33].

Another characteristic arising from man-made landscape changes is reduced structural connectivity [34], which in turn can limit the movement of individuals and, consequently, the gene flow of populations [35]. Interaction between these two (ecological and genetic) processes thus results in population structuring [36]. Consequently, landscape modifications can cause genetic variations within populations [37] and even reduce their variability [38].

As mentioned earlier, recent studies carried out in areas of the Atlantic Forest have shed light on the influence of forest fragmentation on the dispersion and circulation of YFV [32,33]. Therefore, it is essential to verify the role of forest fragmentation also in the populations of the vectors, especially in the species *Hg. leucocelaenus*, in view of its ecological plasticity to impacted environments.

In light of this, the present study sought to identify population structuring and genetic variability in *Hg. leucocelaenus* based on specimens collected in Atlantic Forest fragments in different municipalities in the state of São Paulo. Our hypotheses assume different values of genetic diversity associated with land use in the locations sampled, as well as a possible pattern of population structuring associated with the degree of isolation of the fragments (isolation by geographic distance), i.e., the greater the distance between fragments, the greater the genetic distances among specimens. Likewise, the greater the connectivity of the landscape, the lower the genetic differentiation.

## 2. Materials and Methods

### 2.1. Characterization of the Study Area

The state of São Paulo extends over 248,219.48 km^2^ (24.8 million ha) and has an estimated population of 46,289,333, with a population density of 166.25 inhabitants/km^2^ [39]. Currently, the spatial arrangement in the state is as follows: agricultural sector—17,778,910 ha (71.6% of total land area); area with vegetation—5,577,114 ha (22.5%), of which 3.9 million ha is made up of Atlantic Forest and 163,674 ha of Cerrado; area without vegetation—880,768 ha (3.54%); and water bodies—582,619 ha (2.34%) [40,41] (Figure 1).

### 2.2. Collection of Hg. leucocelaenus Specimens

*Hg. leucocelaenus* specimens were collected in eleven locations in fragments of the Atlantic Forest biome in eight municipalities in the state of São Paulo (Figure 1, Table 1). From this point on, for the sake of readability, the municipality of São José do Rio Preto will be referred to as Rio Preto in the text, tables and figures. Collections took place from November 2018 to December 2020 during the hottest, most humid months (spring and summer) in the daytime from 8:00 a.m. to 4:00 p.m., giving a total of 24 visits to the collection sites. All visits were carried out punctually; thus, the sites could not be visited simultaneously.

Adult individuals were captured at ground level with a dip net and mouth aspirator, while immature forms were collected with wash bottles with flexible silicon nozzles, which were used to drain the contents of the breeding sites found in tree hollows. The immature forms were then sent to the Entomology Laboratory of the School of Public Health at the University of São Paulo, LESP/FSP/USP, where they were kept until the winged forms emerged. The specimens were morphologically identified with taxonomic keys found in the specialized literature [19,42] and then kept at −20 °C until DNA was extracted. Subsequently, the phenotypic identifications were validated using the reference mitochondrial genome (Appendix A).

### 2.3. Sample Preparation and Genotyping by Sequencing (GBS)

Total genomic DNA was extracted from individual mosquitoes with the DNeasy^®^ Blood and Tissue Kit (Qiagen Ltd., Crawley, UK), following the manufacturer’s protocol, and quantification was performed by fluorometry with the QuBit dsDNA HS Assay Kit (ThermoFisher Scientific, Dietikon, Switzerland), according to the manufacturer’s protocol.

In this study, we employed a low-density genome Illumina sequencing protocol [43,44] to obtain whole-mitogenome sequences of *Hg. leucocelaenus.* For this, DNA libraries were prepared using one-fifth of the total volume recommended for the Nextera XT Library prep kit (Illumina); after this modification, the full preparation was carried out, according to the manufacturer’s protocol. After being grouped and loaded into two mid-output flow cells, the DNA samples were sequenced in the NextSeq500 (Illumina) platform using 151 single-read cycles.

FastQC was used to assess the quality of the sequence reads [45], and the processing and quality control of the reads was performed with Trimmomatic [46], with the following filters: removal of adapters if identified; removal of bases with quality below Q20 (Phred Score); removal of reads shorter than 40 bp; and trimming of the first 10 bases sequenced. Total DNA sequencing data were aligned with the reference mitochondrial genome of *Hg. leucocelaenus*—NC_057212.1 (NCBI—www.ncbi.nlm.nih.gov/nuccore/NC_057212.1, accessed on 10 May 2023) using Burrows–Wheeler Aligner 0.7.17 [47]. This option was chosen because there is to date no reference nuclear genome for the species being studied.

Variant calling was performed with bcftools in the SAMtools 1.10 package [48], and the variant panel was exported in VCF (Variant Call Format) version 4.2 format. The variants were filtered with LCVCFtools [49] with the following parameters: allelic frequency of the lowest allele (LAF) ≥ 1%; genotype quality (GQ) ≥ 20; minimum sequencing depth (SD) ≥ 5; and minimum of 75% non-missing data.

### 2.4. Population Diversity and Stratification Analysis

The values for nucleotide diversity within the populations and the analysis of molecular variance (AMOVA) (used to verify the genetic variation within and between groups of individuals, based on the similarity obtained by calculating the sum of squares) for the specimens grouped by collection location were estimated with Arlequin 3.5 [50]. For nucleotide diversity, the values of theta (θ): S and $\hat{\pi }$, indexes which represent the variations within and between the populations, were calculated. The former is estimated from the observed number of polymorphic loci, while the latter is estimated from the mean number of pairwise differences between DNA sequences [51,52], both used to measure the degree of polymorphism present in DNA.

Signs of demographic events in the populations (expansion or retraction) were identified with Tajima’s D test for neutrality [52] and Fu’s FS test [53], indexes that provide a history of the demographic events of the populations (negative values show recent events of population expansion, while positive values indicate possible bottlenecks), while signs of stratification in the populations were investigated with the parameter *F_ST_*, following the mathematical model of Weir and Cockerham [54]. This parameter estimates the probability of two alleles randomly chosen between a population and subdivisions of the population being equal [55], and allows pairwise comparison by municipality sampled. Arlequin was used to estimate the significance (*p* ≤ 0.05) of the mean values of *F_ST_* based on the permutation test (1000 replicates).

Data from variant calling were manipulated with the *vcfR* package [56] to allow the Euclidian distances between the allele frequencies found for the mitochondrial SNPs to be calculated. These were subsequently used to identify signs of stratification among the populations by multidimensional scaling (MDS) and to build hierarchical clusters (1000 replicates) with the aid of two packages: stats [57] and *pvclust* [58], respectively.

The *adegenet* package was used to perform discriminant analysis of principal components (DAPC) among the samples [59]. The eigenvalues were obtained and used to identify the number of genetic clusters (K) that best represents the dataset. Panels identifying the probability of an individual belonging to a particular group, i.e., the degree of similarity among clusters, were produced. Both the clusters and the panels were obtained with the *adegenet* [59] and *reshape2* [60] packages, and their visualizations were produced with *ggplot2* [61] and *ggpubr 0.40* [62]. All the packages referred to above were developed for the R programming language [57].

### 2.5. Landscape Descriptors

Two buffers with radii of 2850 m and 5700 m, respectively, were defined around each collection point to calculate the landscape metrics. The buffer radius was based on an estimate of the expected dispersal radius of 0.3 to 0.5 times the maximum dispersal distance of a species [63]. In the case of *Hg. leucocelaenus*, this corresponds to 5.7 km [64]. Each of the points was georeferenced and plotted on a map of land use in the state of São Paulo (orthophoto mosaic created by supervised classification with RapidEye images from 2013, scale 1:10,000) produced by the Brazilian Foundation for Sustainable Development (FBDS) (www.fbds.org.br/, accessed on 10 May 2023) (Figure 1). This was used to obtain landscape elements according to the FBDS land-use classification: (1) Forest Formation—Tree vegetation native to the Atlantic Forest with a continuous canopy; (2) Water—Continuous water surface; (3) Human-impacted Area—Areas without any native vegetation cover; (4) Built Area—Areas with buildings (www.ibge.gov.br/geociencias/cartas-e-mapas/bases-cartograficas-continuas/15759-brasil.html, accessed on 10 May 2023), (5) Non-forest Formation—Native bushy or herbaceous vegetation; and (6) Forestry—*Eucalyptus* sp. or *Pinus* sp. plantations.

Subsequently, the land-use classes were used to quantify the landscape metrics: Total area (CA) and Percentage of the landscape occupied (PLAND), and the Total Edge (TE) and Density Edge (DE) metrics were also calculated for the Forest Formation class. Regression analysis was performed, using the landscape metrics as the predictor variable and the values for the diversity indices theta S and theta $\hat{\pi }$ as the response variables, to identify any possible influence of the land-use classes on genetic diversity patterns in the sampled populations.

## 3. Results

### 3.1. Genetic Diversity

After sequencing, the average depth for each specimen sequenced was approximately 1,300,000 reads, representing an average coverage of 1.23× for the full genome, as expected for a low-density sequencing protocol.

Because the complete genome of the species *Hg. leucocelaenus* has not been sequenced to date, we had to use the mitochondrial genome (NC_057212.1). A total of 115 mitochondrial SNPs were genotyped, corresponding to a genotyping rate of 51.11% (Appendix A).

When all the SNPs (115) were analyzed, the sampled populations were found to have on average 23 loci, with less than 25% of missing data (sites with any of the four nucleotides C, T, A and G). Pindamonhangaba had the greatest number of these loci [50], while Miracatu had the smallest number (3). As for the polymorphism in the loci analyzed, once again, Pindamonhangaba had the highest value [31], followed by the municipalities of Rio Preto and Balsamo, which had 24 and 17, respectively. These three populations also had the highest values for the θ diversity indices: Pindamonhangaba (θS = 10.96, θ$\hat{\pi }$ = 9.53), Balsamo (θS = 6.94, θ$\hat{\pi }$ = 6.05) and Rio Preto (θS = 6.50, θ$\hat{\pi }$ = 6.26). In contrast, Miracatu did not exhibit any genetic variability (θS = 0, θ$\hat{\pi }$ = 0) and was followed by São Paulo (θS = 1.31, θ$\hat{\pi }$ = 0.56), Igaratá (θS = 2, θ$\hat{\pi }$ = 2), Mairiporã (θS = 2.44, θ$\hat{\pi }$ = 1.66) and Caraguatatuba (θS = 2.69, θ$\hat{\pi }$ = 2.21) (Appendix A).

### 3.2. Neutrality Test

In the neutrality test to identify signs of population expansion or retraction, only São Paulo and Mairiporã exhibited concordance in the statistically significant results for Tajima’s D and Fu’s FS tests; both had negative values, suggesting expansion events. For the other populations, there were no statistically significant results for Tajima’s D test, while for Fu’s FS test, statistically significant negative values were observed for the Rio Preto, Caraguatatuba, Balsamo and Pindamonhangaba populations (Table 2).

### 3.3. Genetic Structure

In the pairwise comparisons between the populations, the largest value of *F_ST_* was recorded between the municipalities of Igaratá and Miracatu (1), followed by Balsamo and São Paulo (0.74982), Balsamo and Mairiporã (0.67021), São Paulo and Rio Preto (0.6422) and Balsamo and Igaratá (0.62904). Generally, the two populations in the São José do Rio Preto mesoregion (Rio Preto and Balsamo) had the largest values in most of the comparisons with the other municipalities sampled, and the smallest value for these two populations was precisely when they were compared with each other (0.1099). The smallest value for all the comparisons was recorded between São Paulo and Mairiporã (0.0201) (Appendix A).

Overall, there was concordance between the values observed for the *F_ST_* index and the geographic distances between the collection sites. Geographic distance not only has a strong association with the pattern of genetic structure observed, but also explains by itself around 45% of the variation in the data, suggesting clear isolation by distance (r^2^: 0.4509; *p*-value < 0.001 [Mantel test; *p*-value < 0.01; 9999 replicates], Table 3, Appendix A). The distances between the collection sites vary from 5 to 535 km: the populations in the São José do Rio Preto mesoregion are less than 30 km apart, and the distances from these populations to the others vary between 400 and 535 km, while the distances between the populations in the east of the state range from 5 to 230 km.

Multidimensional scaling revealed greater similarity among the specimens sampled from the northwest of the state of São Paulo (municipalities of Balsamo and Rio Preto). These are closer to each other in the bidimensional space than to the populations in the eastern region of the state, and Caraguatatuba is segregated from the remainder, which are close to each other (Appendix A). This pattern was corroborated by the hierarchical cluster analysis, which shows that the specimens in Balsamo and Rio Preto are clearly segregated, with the other specimens in a separate clade, in which the Caraguatatuba population is the outermost branch and the remaining populations are organized as follows: a more external branch formed by Miracatu, and two branches formed by Pindamonhangaba and São Paulo, and Igaratá and Mairiporã (Figure 2).

Discriminant analysis of principal components once again revealed overlapping of the specimens in the northwest of the state of São Paulo, with a few individuals in the Pindamonhangaba and São Paulo populations showing greater similarity with individuals from Balsamo and Rio Preto. However, most of the specimens in the populations, other than those from Balsamo and Rio Preto, overlap to form practically a single cluster (Appendix A). With the results obtained using the eigenvalues, three groups (K = 3) were found to best represent the dataset: the populations in the northwest of the state (Balsamo and Rio Preto) form two distinct genetic groups, and the other practically homogeneous populations form a single group (Appendix A). Comparison of the gene signature pattern of the individuals showed a clear correspondence between the individuals collected in Balsamo and Rio Preto, while the others once again formed a practically homogeneous group (Figure 3).

### 3.4. Landscape Metrics and Generalized Linear Models

When the landscape metrics were compared for the 2850 m and 5700 m buffers, the locations with the greatest genetic diversity were found to have a high proportion of agricultural land use, at least in the municipalities of Balsamo (2850 m buffer—79.7%; 5700 m—76.9%) and Rio Preto (2850 m buffer—92.9%; 5700 m buffer—84.9%), whereas the corresponding figures for Pindamonhangaba were 32% and 50.1%, respectively (Appendix A). Interestingly, when the influences of the various land-use classes on the observed diversity indices were compared with the aid of generalized linear models, the agricultural land-use class was the only one that had an association with the observed diversity pattern, particularly when the influence of this class was determined based on a 5700 m buffer around the collection points. The area used for agriculture had a better fit for the theta $\hat{\pi }$ index (slope = 0.07598, *p*-value = 0.03417) and accounted for approximately 55% of the observed pattern (M2pi_5700m), while for the theta S index, this area had a less accurate fit (slope = 0.07723, *p*-value = 0.05826) and accounted for approximately 48% of the observed pattern (M2S_5700m) (Table 4 and Table 5).

Another characteristic that the municipalities with the highest values for the genetic diversity indices had in common was the low proportion of forest cover. Balsamo and Rio Preto had the lowest values (2850 m buffer—13.8% and 5700 m buffer—12.5% and 2850 m buffer—4.3% and 5700 m buffer—5.8%, respectively), while the corresponding figures for Pindamonhangaba were 66% and 47.6%, respectively (Appendix A). However, no association between the Forest class and the observed pattern could be identified (Table 4 and Table 5, Appendix A). The values for forest edge (in linear meters) for the three municipalities varied between 65,000 and 161,000 m (2850 m buffer) and 267,000 and 601,000 m (5700 m buffer), with a density ranging from 10 to 63 m/ha (2850 m buffer) and 16 to 58 m/ha (5700 m buffer) (Appendix A).

## 4. Discussion

This study tested the hypothesis that genetic diversity is associated with land use in the areas where samples were collected and that population structuring of *Hg. leucocelaenus* may be occurring as a function of the degree of isolation of the sampled vegetation fragments (isolation by distance) in municipalities in the state of São Paulo. The results show greater genetic diversity (theta S and $\hat{\pi }$ indices) among the populations from Pindamonhangaba, Rio Preto and Balsamo, the latter two exhibiting clear structuring compared with the other populations sampled. The populations from Rio Preto and Balsamo, which are located in the northeast of the state in the São José do Rio Preto mesoregion, constituted a separate clade, while the other populations exhibited greater similarity to each other, although Caraguatatuba formed a distant branch.

The populations with the greatest genetic diversity indices (number of polymorphic loci and mean pairwise differences between the sequences) were found in areas with a high degree of agricultural land use (>32%). When only the populations in the São José do Rio Preto mesoregion were considered, agricultural land use exceeded 75% of the areas sampled, clearly showing there is a strong association between this land-use class and the pattern of genetic diversity in these populations, as shown by the theta $\hat{\pi }$ index.

Landscape structure, i.e., differences in land use, are directly associated with different patterns in mosquito communities [65], whose richness and diversity vary according to the species’ response to environmental changes [66]. At species level, these changes can lead to microevolutionary events, i.e., changes in the genetic structure of populations in a short period, reflecting the way species react to the environment [67]. For example, in the particular case of the genus *Haemagogus*, Mucci et al. [68] found that the species *Hg. janthinomys*/*capricornii* had multiple blood meals (on birds, cattle and primates), indicating the influence of areas adjacent to forest fragments on the feeding habits of this genus. Curiously, the same study failed to identify multiple blood meals for the species *Hg. leucocelaenus*, which has greater ecological plasticity than *Hg. janthinomys*/*capricornii* [69], 2010). In addition, exposure to pathogens can trigger microevolutionary events in mosquitoes and sometimes affect their vectorial capacity [70,71].

The results of the present study may, therefore, reflect not only the plasticity of *Hg. leucocelaenus*, specimens of which are found throughout the São José do Rio Preto region [72], but also the pressure imposed by YFV. There have been reports of this virus circulating in the region over the last decades [73], where the virus has been isolated from sylvatic mosquitoes [74]. However, it should be noted that unlike the populations mentioned above, the population with the greatest values of genetic diversity (Pindamonhangaba) is in a region where there had been no record of epizootics in NHPs until 2018, when an epizootic was confirmed [74,75,76].

Regarding demographic events (population expansion and retraction), for six out of eight, or the majority of the populations, the results of at least one of the neutrality tests were negative and statistically significant, but only the results for the São Paulo and Mairiporã populations were concordant between the two tests (Tajima’s D and Fu’s FS). Negative values in these tests generally indicate recent population expansion. Common to the areas where specimens of these two populations were collected are the following factors: (i) they are located in Cantareira State Park (CSP), an important remnant of Atlantic Forest in the state of São Paulo and considered the largest urban forest in the world [77]; (ii) they have the largest proportion of built-up areas; and (iii) they have a large forest edge surface (between 109,000 and 321,000 m), with a density of between 31 and 43 m/ha.

The juxtaposition of urban areas and forests has a direct impact on plant species composition [78], which can lead to a reduction in the number of typical breeding sites (tree hollows) of the species *Hg. leucocelaenus* [79]. However, previous studies in which surveys of mosquito fauna were carried out had already recorded the presence of *Hg. leucocelaenus* in areas close to the edge of CSP and in locations with different degrees of plant cover [28,80,81]. Once again, the results of the present study may be reflecting the ecological plasticity of this species to environments with a greater degree of man-made changes [69], which could in turn be contributing to the population expansion events observed in the two populations sampled inside CSP.

Although pairwise comparison of the populations yielded high values of *F_ST_* (0.02–0.75), these agree with the findings reported by Alonso et al. [44] in a study of populations of *Mansonia* spp., which used the same protocol to identify mitochondrial SNPs as in our study. In terms of the degree of population structuring, the specimens collected in the northwest of the state of São Paulo are clearly separated from the populations in the eastern region of the state and also have the largest values of *F_ST_* in the pairwise comparison with these populations. The smaller values for the populations outside the northwest region indicate greater genetic proximity among these populations and suggest that if there is a physical barrier, this is not a complete obstacle to the dispersal of individuals. In this regard, our results corroborate the idea of isolation by distance, according to which the greater the distance between species, the greater the genetic differentiation observed. In a slightly larger-scale study with populations of Aedes scapularis, another sylvatic species, Petersen et al. [82] observed a correspondence between the genetic and geographic distances of the populations sampled. In contrast, previous studies with mosquitoes in the state of São Paulo revealed weak genetic structuring and the absence of isolation by distance both for sylvatic species (*Aedes fluviatilis* and *Anopheles cruzii*) and an invasive species (*Ae. aegypti*) [83,84,85]. Nevertheless, these studies were carried out on a microgeographic scale (only in the city of São Paulo), whereas the present study extended over hundreds of kilometers.

Mosquitoes generally have a dispersal range of around 6 km [86]. For the genus *Haemagogus*, distances of between 5 and 11 km have been recorded [64], and in the specific case of *Hg. leucocelaenus*, a distance of 5 km was recorded. However, mosquitoes can cover large distances by passive dispersal [86], and in some cases, these distances may reach hundreds of kilometers [87]. The greatest geographic distances between sampling points in our study are those between the municipalities in the northwest of the state and those in the eastern region (>400 km), but there are still substantial distances between the municipalities outside the northwest (up to 230 km). Even if it were assumed that the specimens were able to cover large geographic distances by passive dispersal, it would be unreasonable to attribute to this variable alone the degree of genetic similarity observed between these specimens.

Likewise, the populations that formed a single clade (the Vale do Ribeira, Vale do Paraíba and Metropolitan mesoregions) and that are separated by distances of between 5 and 230 km are part of, or are located close to, a continuous strip of Atlantic Forest remnant, a large fragment of vegetation extending along the Atlantic coast of the state of São Paulo [88]. This region may be an area in which *Hg. leucocelaenus* specimens can travel along the so-called “ecological corridors” found in the vegetation fragments, like those through which YFV spread in recent epizootics [89]. In addition, the urban form itself, with, for example, highways close to forest edges, can contribute to the displacement of this species, as observed for the dispersal of YFV by Prist et al. [32]. It is, therefore, reasonable to suppose that the pattern observed here can be explained by dispersal of *Hg. leucocelaenus* aided by a certain degree of landscape connectivity.

The type of molecular marker used in the present study (mitochondrial SNPs) has been successfully used for population characterization both on a macro [90,91] and micro [44] scale, showing its suitability for use in studies of population genetics. Mitochondrial SNPs are, thus, an excellent alternative for studying population genetics when the genome of the target species has not been completely sequenced. To our knowledge, the present study is the first to use this marker for the species *Hg. leucocelaenus*.

A possible limitation of this study concerns the collection of immature specimens, in particular, in the municipality of Miracatu, where a single collection was performed. Consequently, there is a large chance that the individuals collected constitute the same brood and, therefore, that our results for this area do not accurately reflect the genetic variability of this population.

## 5. Conclusions

Greater genetic diversity in *Hg. leucocelaenus* mosquitoes was associated with greater agricultural land use in the areas sampled, and signs of population expansion for this species were observed in areas with the largest values of forest edge surface. In addition, clear population structuring associated with the distance separating the fragments sampled (isolation by distance) was observed, and the populations from a large area of vegetation (a large fragment of the Atlantic Forest that extends along the Atlantic coast of the state of São Paulo) exhibited greater similarity to each other.

## Figures and Tables

**Figure 1 genes-14-01671-f001:**
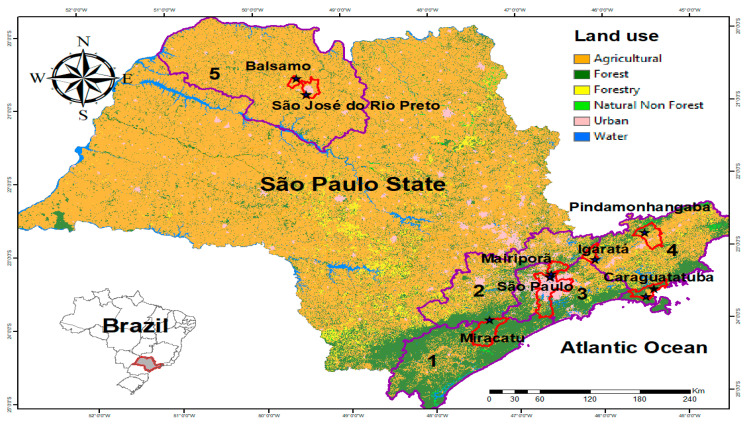
Map of the state of São Paulo with the study areas highlighted. Mesoregions are shown in purple outline. Southern Coast (1); Macrometropolitan, made up of 36 municipalities in four microregions (2); Metropolitan (3); Paraíba Valley (4); and São José do Rio Preto (5). The municipalities where specimens were collected are shown in red outline, and the collection sites are represented by black stars. The map was created in ArcGis version 10.2 (resources.arcgis.com/en/help/main/10.2/index.html, accessed on 10 May 2023).

**Figure 2 genes-14-01671-f002:**
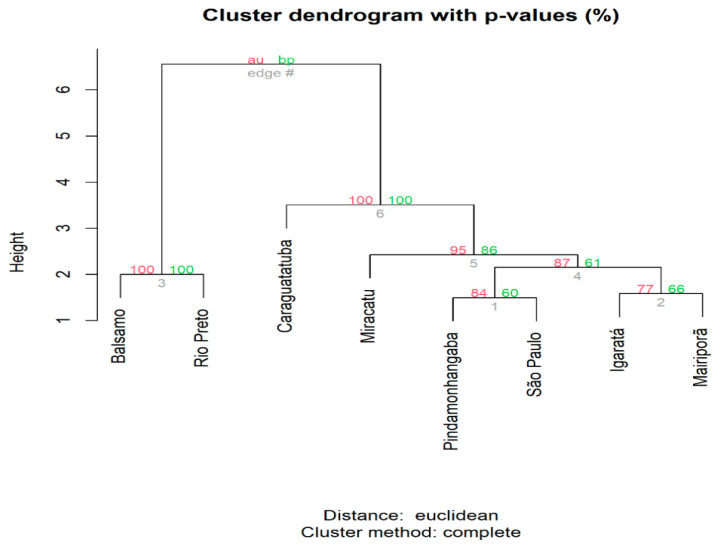
Dendrogram showing the clusters formed by the sampled populations based on the Euclidean distances between the allele frequencies found for the mitochondrial SNPs (1000 replicates). AU: approximately unbiased; BP: bootstrap probability.

**Figure 3 genes-14-01671-f003:**
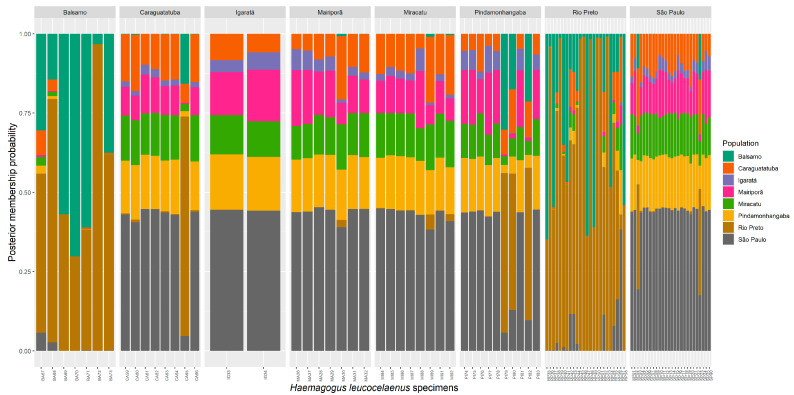
Probability of individuals belonging to a particular group rather than to the original population they were collected from. Individuals are identified by vertical lines. The length of the segments represents the probability of the individual belonging to a particular genetic group rather than to the original population it came from.

**Table 1 genes-14-01671-t001:** Locations where adult specimens and immature forms (in parentheses) of *Hg. leucocelaenus* were collected to analyze single nucleotide polymorphisms (SNPs).

Municipality	Location	Description	Geographical Coordinates	Number of Specimens			Total
				♀	♂	*	
Balsamo	Sítio Madalena	Private rural property	20°39′46.199″ S 49°31′20.302″ W	7			7
Caraguatatuba	Sítio Marisquinho	Private rural property	23°43′23.099″ S 45°30′41.198″ W	3	(1)		3(1)
	Benfica district	Private rural property	23°36′51.998″ S 45°25′0.998″ W	4			4
Igaratá	Pontal das Garças Gated Community	Private property	23°12′41.000″ S 46°6 21.899″ W	2			2
Mairiporã	Cantareira State Park	Pinheirinho trail	23°24′27.691″ S 46°37′11.010″ W	6(1)			6(1)
Miracatu	“Legado das Águas” Ecological Reserve	Suspended trail	24°1′54.199″ S 47°21 10.699″	(4)	(1)	(3)	(8)
Rio Preto	Macacos Woods	Vegetation fragment south of the dirt road cutting this fragment	20°53′4.301″ S 49°24′58.201″ W	11			11
		Vegetation fragment north of the dirt road cutting this fragment	20°53′4.898″ S 49°24′51.599″ W	13			13
São Paulo	Cantareira State Park	Bica trail	23°27′8.600″ S 46°38′8.700″ W	6(2)	2(2)		8(4)
		Administration sector	23°26′49.992″ S 46°37′57.767″ W	3	(11)		3(11)
Pindamonhangaba	Trabiju Municipal Park	“Water tank” trail	22°50′23.201″ S 45°31′24.100″ W	10			10
Total				68(7)	2(12)	(3)	70(22)

* Immature forms that did not complete the developmental life cycle.

**Table 2 genes-14-01671-t002:** Neutrality test based on the sampled *Hg. leucocelaenus* populations.

Neutrality Test	Statistics	SP	MA	IG	RP	CA	BA	PI	MI
Tajima’s D test	Tajima’s D	−1.60937	−1.62291	0	−0.13906	−0.85567	−0.7151	−0.62518	0
	Tajima’s D *p*-value	0.026	0.003	1	0.535	0.222	0.265	0.308	1
Fu’s FS test	FS	−3.40282 × 10^38^	−6.42258	0.6932	−21.3184	−6.82143	−2.567	−3.6223	1 × 10^−6^
	FS *p*-value	<0.001	<0.001	0.367	<0.001	<0.001	0.031	0.021	1

SP—São Paulo; MA—Mairiporã; IG—Igaratá; RP—Rio Preto; CA—Caraguatatuba; BA—Balsamo; PI—Pindamonhangaba; MI—Miracatu.

**Table 3 genes-14-01671-t003:** Linear model using as predictor variable the geographic distances between the collection locations and as response variable the values of $F_{ST}$ from the pairwise comparisons between the populations.

Predictor Variable	Intercept	Slope	SE	t-Value	Pr(>|z|)	r^2^	F-Statistic	DF	*p*-Value
Distance	−1.87		0.5888	−3.176	0.003944 **	0.4509	20.53	25	0.000126
		0.5081	0.1121	4.531	0.000126 ***				

SE—Standard error; DF—Degrees of freedom; ** significant at *p* < 0.01 level; *** significant at *p* < 0.001 level.

**Table 4 genes-14-01671-t004:** Generalized linear models using land-use class as the predictor variable and diversity index theta S as the dependent variable.

Models	Dependent Variable	Predictor Variable	Intercept	Slope	SE	t-Value	Pr(>|z|)	r^2^	DF
M1S_2850m	theta S	Water	4.59829		1.46422	3.14	0.0201 *	0.09843	6
				−0.08244	0.10185	−0.809	0.4492		
M2S_2850m	theta S	Agricultural use	1.99924		1.68306	1.188	0.28	0.329	6
				0.06009	0.03504	1.715	0.137		
M3S_2850m	theta S	Urban	5.04073		1.52613	3.303	0.0163 *	0.1721	6
				−0.11166	0.09997	−1.117	0.3067		
M4S_2850m	theta S	Forest	5.87074		2.44473	2.401	0.0532	0.1093	6
				−0.03582	0.04173	−0.858	0.4237		
M5S_2850m	theta S	Forestry	2.8586		1.4329	1.995	0.0931	0.2864	6
				0.9119	0.5877	1.552	0.1717		
M6S_2850m	theta S	ED	1.71488		3.14653	0.545	0.605	0.1049	6
				0.06902	0.0823	0.839	0.434		
M7S_2850m	theta S	TE	1.70 × 10^0^		3.12 × 10^0^	0.545	0.605	0.108	6
				2.74 × 10^−5^	3.21 × 10^−5^	0.852	0.427		
M1S_5700m	theta S	Water	4.5354		1.4739	3.077	0.0217 *	0.0787	6
				−0.1162	0.1623	−0.716	0.501		
M2S_5700m	theta S	Agricultural use	1.15875		1.62083	0.715	0.5015	0.476	6
				0.07723	0.03308	2.335	0.0583		
M3S_5700m	theta S	Urban	4.92998		1.55404	3.172	0.0193 *	0.1364	6
				−0.07013	0.07203	−0.974	0.3678		
M4S_5700m	theta S	Forest	6.38406		2.22982	2.863	0.0287 *	0.2026	6
				−0.05156	0.04177	−1.235	0.2632		
M5S_5700m	theta S	Non-forest formation	4.629		1.508	3.07	0.0219 *	0.08719	6
				−860.578	1136.74	−0.757	0.4777		
M6S_5700m	theta S	Forestry	3.2103		1.5494	2.072	0.0837	0.1544	6
				0.3985	0.3808	1.047	0.3356		
M7S_5700m	theta S	ED	1.643		2.76258	0.595	0.574	0.1454	6
				0.07531	0.07454	1.01	0.351		
M8S_5700m	theta S	TE	1.66 × 10^0^		2.73 × 10^0^	0.61	0.564	0.1477	6
				7.38 × 10^−6^	7.24 × 10^−6^	1.02	0.347		
Null Model	theta S		4.107		1.30 × 10^0^	3.16 × 10^0^	0.0159 *		7

SE—Standard error; DF—Degrees of freedom; * significant at the *p* < 0.05 level.

**Table 5 genes-14-01671-t005:** Generalized linear models using land-use class as the predictor variable and diversity index theta $\hat{\pi }$ as the dependent variable. * significant at the *p* < 0.05 level.

Models	Dependent Variable	Predictor Variable	Intercept	Slope	SE	t-Value	Pr(>|z|)	r^2^	DF
M1pi_2850m	$\mathrm{theta}\;\hat{\pi }$	Water	3.91061		1.35692	2.882	0.028 *	0.06887	6
				−0.06288	0.09439	−0.666	0.53		
M2pi_2850m	theta $\hat{\pi }$	Agricultural use	1.41377		1.45007	0.975	0.3672	0.401	6
				0.0605	0.03019	2.004	0.0919		
M3pi_2850m	theta $\hat{\pi }$	Urban	4.52117		1.34167	3.37	0.015 *	0.2306	6
				−0.11784	0.08788	−1.341	0.229		
M4pi_2850m	theta $\hat{\pi }$	Forest	5.41905		2.17796	2.488	0.0473 *	0.1499	6
				−0.03824	0.03718	−1.029	0.3433		
M5pi_2850m	theta $\hat{\pi }$	Forestry	2.3998		1.3077	1.835	0.116	0.2852	6
				0.8298	0.5363	1.547	0.173		
M6pi_2850m	theta $\hat{\pi }$	ED	1.86668		2.93816	0.635	0.549	0.06144	6
				0.04816	0.07685	0.627	0.554		
M7pi_2850m	theta $\hat{\pi }$	TE	1.85 × 10^0^		2.91 × 10^0^	0.634	0.55	0.06397	6
				3.00 × 10^−5^	3.00 × 10^−5^	0.64	0.546		
M1pi_5700m	$\mathrm{theta}\;\hat{\pi }$	Water	3.85752		1.36241	2.831	0.0299 *	0.05335	6
				−0.08723	0.15001	−0.581	0.5821		
M2pi_5700m	$\mathrm{theta}\;\hat{\pi }$	Agricultural use	0.63527		1.36349	0.466	0.6577	0.5541	6
				0.07598	0.02783	2.73	0.0342 *		
M3pi_5700m	$\mathrm{theta}\;\hat{\pi }$	Urban	4.40785		1.37735	3.2	0.0186 *	0.1842	6
				−0.07431	0.06384	−1.164	0.2886		
M4pi_5700m	$\mathrm{theta}\;\hat{\pi }$	Forest	5.77105		1.99192	2.897	0.0274 *	0.2347	6
				−0.05062	0.03731	−1.357	0.2237		
M5pi_5700m	$\mathrm{theta}\;\hat{\pi }$	Non-forest formation	4.111		1.344	3.058	0.0223 *	0.1272	6
				947.928	1013.598	−0.935	0.3858		
M6pi_5700m	$\mathrm{theta}\;\hat{\pi }$	Forestry	2.6975		1.4062	1.918	0.104	0.1622	6
				0.3726	0.3456	1.078	0.322		
M7pi_5700m	$\mathrm{theta}\;\hat{\pi }$	ED	1.66811		2.58455	0.645	0.543	0.1005	6
				0.05709	0.06973	0.819	0.444		
M8pi_5700m	$\mathrm{theta}\;\hat{\pi }$	TE	1.68 × 10^0^		2.55 × 10^0^	0.657	0.535	0.1029	6
				5.61 × 10^−6^	6.77 × 10^−6^	0.829	0.439		
Null Model	$\mathrm{theta}\;\hat{\pi }$		3.536		1.19 × 10^0^	2.984	0.0204 *		7

## Data Availability

Not applicable.

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
