# Peer review of "Genetic Structuring of One of the Main Vectors of Sylvatic Yellow Fever: Haemagogus (Conopostegus) leucocelaenus (Diptera: Culicidae)"

_genes, 2023, doi:10.3390/genes14091671_

Round 1
Reviewer 1 Report
The authors studied mitochondrial genetic structure of the mosquito species Haemagogus leucocelaenus in the state of São Paulo, Brazil. They revealed potential recent expansion events and investigated the relationship between the population diversity and landscape structure. It is impressive to see diverse methods had been used to support the result. However, there are some missing points that remain to be further addressed and I hope the following considerations can help improve the manuscript:
Major:
The authors focused on studying the mitochondrial genome rather than the nuclear genome. But the distinction was not clearly stated, for example: the title, in line 343, the introduction and methods. It is worth noting that mitochondrial genome is maternally inherited, the authors should set it apart from nuclear genome analysis.
In methods, it is a lack of clarity regarding the sample processing. The manuscript mentions a two-year collection period, but it remains unclear whether the authors collected samples from different locations simultaneously or at different time intervals. It would be helpful to specify the number of samples collected at each instance to better understand the study's temporal and spatial aspects. Furthermore, the methods section does not provide sufficient information on how the authors identified the species. It is important to clarify whether the identification was based on phenotype (morphological characteristics) or genotype (genetic markers). Since this study focuses on the mitochondrial genome, the authors should justify that the samples analyzed indeed belong to the targeted species. Moreover, the manuscript lacks details about the sequencing design, including sequence depth and coverage for each individual and population. The authors should provide information about the sequencing depth to assess the reliability of the results, and whether only mitochondria were extracted and sequenced. As in the result, the coverage of the data appears to be relatively low, as indicated in lines 571-573, and this should be addressed and discussed to better interpret the findings.
Minor:
Line 569, it is still possible to study nuclear genome (De nova sequencing)
Figure 1, The collection sites for 2 and 3 are not clear. Are they very close to one another? If so, what causes their genotypes to be segregated according to the Dendrogram?
Table 1, what does the bracket mean?
Author Response
Dear Reviewer
Thank you very much for reviewing our manuscript and for the suggestions, which have enabled us to greatly improve the text.
We have made the changes to the manuscript to address the points, and we hope the article is now suitable for publication in Insects.
Sincerely,
Mauro Marrelli
Reviewer 1
Comments and Suggestions for Authors
The authors studied mitochondrial genetic structure of the mosquito species Haemagogus leucocelaenus in the state of São Paulo, Brazil. They revealed potential recent expansion events and investigated the relationship between the population diversity and landscape structure. It is impressive to see diverse methods had been used to support the result. However, there are some missing points that remain to be further addressed and I hope the following considerations can help improve the manuscript:
We thank the reviewer for their time and comments which have greatly strengthened the manuscript. Below is an enumerated response to each of the comments.
Major:
The authors focused on studying the mitochondrial genome rather than the nuclear genome. But the distinction was not clearly stated, for example: the title, in line 343, the introduction and methods. It is worth noting that mitochondrial genome is maternally inherited, the authors should set it apart from nuclear genome analysis.
In methods, it is a lack of clarity regarding the sample processing. The manuscript mentions a two-year collection period, but it remains unclear whether the authors collected samples from different locations simultaneously or at different time intervals. It would be helpful to specify the number of samples collected at each instance to better understand the study's temporal and spatial aspects.
Response: We thank you for the comment. We agree that there was a lack of clarity regarding the sample processing. The information was included in the text (lines 488-489), and a new supplementary table (Supplementary Table 4) containing the number of samples collected at each collection dates was added.
“Supplementary Table 4 is at the end of this document"
Furthermore, the methods section does not provide sufficient information on how the authors identified the species. It is important to clarify whether the identification was based on phenotype (morphological characteristics) or genotype (genetic markers). Since this study focuses on the mitochondrial genome, the authors should justify that the samples analyzed indeed belong to the targeted species.
Response: We thank you for the comment. Modification and additional information were included in the text (Lines 496 – 499).
“The specimens were morphologically identified with taxonomic keys found in the specialized literature [19,42] and then kept at -20°C until DNA was extracted. Subsequently, the phenotypic identifications were validated using the reference mitochondrial genome.”
Moreover, the manuscript lacks details about the sequencing design, including sequence depth and coverage for each individual and population. The authors should provide information about the sequencing depth to assess the reliability of the results, and whether only mitochondria were extracted and sequenced. As in the result, the coverage of the data appears to be relatively low, as indicated in lines 571-573, and this should be addressed and discussed to better interpret the findings.
Response: Thank you for your comment. We have provided more information about sequencing design in the text of the MS.
The rapid development of technologies involved in whole-genome sequencing (WGS) has resulted in dramatic reductions in the per base sequencing cost. However, studies that require the sequencing of large numbers of samples remain costly, possibly prohibitively so in some laboratories. One low-cost strategy is genotyping-by-sequencing for low-coverage WGS (L-WGS) defined as around 1x of coverage, which also can be associated with imputation techniques to provide sufficient genomic information to select markers accurately when genotyping rate is too low. The accuracy of variant detection is low in genomes with low coverage depth and tends to have a high false positive rate, but this is attenuated when information between samples is combined, providing good common variant identification power. The inference of genotypes by imputation for both panel-based genotyping and sequencing genotyping has been shown to be accurate, allowing for the potential use of extreme low-coverage (0.2 to 0.8 X of coverage) WGS (EXL-WGS) to discover variants at a dramatic reduction in cost when compared to standard WGS. This was the rationale for using low density or low coverage sequencing protocols in our study, moreover the genotyping rate was sufficient enough (more than 75%) for the 115 SNPs used in our study.
Minor:
Line 569, it is still possible to study nuclear genome (De nova sequencing)
Response: Thank you for your comment. Unfortunately, to date the complete genome of the species Haemagogus leucocelaenus is not yet available.
Figure 1, The collection sites for 2 and 3 are not clear. Are they very close to one another? If so, what causes their genotypes to be segregated according to the Dendrogram?
Response: We thank you for the comment. We agree that the collection sites for 2 and 3 are not clear. Really the two sites despite comprising two municipalities (São Paulo and Mairiporã), have small distance between each other (< 6 kilometers). However, even with this short distance and being inserted within the Cantareira State Park, their areas have distinct characteristics, both in the configuration of the landscape and in the floristic composition, as we observed in a previous work (https://doi.org/10.1016/j.actatropica.2020.105385), which in turn may be contributing to the dissimilarity between the species of the two localities, such as the pattern displayed from the dendrogram.
Table 1, what does the bracket mean?
Response: It represents specimens collected in their immature forms, from the breeding site types of the species (tree hollows).
Table S4. Locations and numbers of adult specimens and immature forms of Haemagogus leucocelaenus collected to analyze single nucleotide polymorphisms (SNPs).
|
Date |
Location |
Description |
Gender |
Form |
|
12/16/2020 |
Balsamo (Sítio Madalena)
|
Private rural property |
F |
Adult |
|
F |
Adult |
|||
|
F |
Adult |
|||
|
F |
Adult |
|||
|
F |
Adult |
|||
|
F |
Adult |
|||
|
F |
Adult |
|||
|
11/3/2019 |
Caraguatatuba (Benfica district) |
Private rural property |
F |
Adult |
|
F |
Adult |
|||
|
F |
Adult |
|||
|
F |
Adult |
|||
|
11/2/2019 |
Caraguatatuba (Sítio Marisquinho) |
Private rural property |
M |
Immature |
|
F |
Adult |
|||
|
F |
Adult |
|||
|
F |
Adult |
|||
|
12/4/2019 |
Igaratá (Pontal das Garças Gated Community) |
Private property |
F |
Adult |
|
F |
Adult |
|||
|
11/26/2018 |
Mairiporã (Cantareira State Park) |
Pinheirinho trail |
F |
Immature |
|
1/8/2019 |
F |
Adult |
||
|
F |
Adult |
|||
|
F |
Adult |
|||
|
F |
Adult |
|||
|
1/27/2020 |
F |
Adult |
||
|
F |
Adult |
|||
|
1/24/2019 |
Miracatu (Legado das Águas” Ecological Reserve) |
Suspended trail |
M |
Immature |
|
M |
Immature |
|||
|
M |
Immature |
|||
|
M |
Immature |
|||
|
F |
Immature |
|||
|
* |
Immature |
|||
|
* |
Immature |
|||
|
* |
Immature |
|||
|
12/10/2019 |
Pindamonhangaba (Trabiju Municipal Park) |
“Water tank” trail |
F |
Adult |
|
F |
Adult |
|||
|
F |
Adult |
|||
|
F |
Adult |
|||
|
F |
Adult |
|||
|
12/11/2019 |
F |
Adult |
||
|
F |
Adult |
|||
|
F |
Adult |
|||
|
F |
Adult |
|||
|
F |
Adult |
|||
|
12/15/2020 |
Rio Preto (Macacos Woods)
|
Vegetation fragment south of the dirt road cutting this fragment |
F |
Adult |
|
F |
Adult |
|||
|
F |
Adult |
|||
|
F |
Adult |
|||
|
F |
Adult |
|||
|
F |
Adult |
|||
|
F |
Adult |
|||
|
F |
Adult |
|||
|
F |
Adult |
|||
|
F |
Adult |
|||
|
F |
Adult |
|||
|
12/18/2020 |
Rio Preto (Macacos Woods)
|
Vegetation fragment north of the dirt road cutting this fragment |
F |
Adult |
|
F |
Adult |
|||
|
F |
Adult |
|||
|
F |
Adult |
|||
|
F |
Adult |
|||
|
F |
Adult |
|||
|
F |
Adult |
|||
|
F |
Adult |
|||
|
F |
Adult |
|||
|
F |
Adult |
|||
|
F |
Adult |
|||
|
F |
Adult |
|||
|
F |
Adult |
|||
|
11/26/2018 |
São Paulo (Cantareira State Park) |
Bica trail |
F |
Immature |
|
M |
Immature |
|||
|
1/15/2019 |
F |
Adult |
||
|
F |
Adult |
|||
|
F |
Adult |
|||
|
F |
Adult |
|||
|
F |
Adult |
|||
|
F |
Adult |
|||
|
9/5/2019 |
M |
Adult |
||
|
M |
Adult |
|||
|
9/26/2019 |
F |
Immature |
||
|
M |
Immature |
|||
|
9/5/2019 |
São Paulo (Cantareira State Park) |
Administration sector |
F |
Immature |
|
M |
Immature |
|||
|
M |
Immature |
|||
|
M |
Immature |
|||
|
M |
Immature |
|||
|
M |
Immature |
|||
|
9/26/2019 |
M |
Immature |
||
|
F |
Immature |
|||
|
12/18/2019 |
F |
Adult |
||
|
F |
Adult |
|||
|
1/15/2020 |
F |
Adult |
||
|
M |
Immature |
|||
|
F |
Immature |
|||
|
M |
Immature |
*Immature forms that did not complete the developmental life cycle

Reviewer 2 Report
Authors investigated the genetic diversity and population structuring of Hg. leucocelaenus mosquitoes in various municipalities in the state of São Paulo. Using mitochondrial SNPs as molecular markers, the study revealed that genetic diversity was associated with agricultural land use, with higher diversity in regions with increased agricultural activity. Population structuring was observed, with populations in the northwest region forming a distinct clade. The results suggest landscape connectivity as a key factor in mosquito dispersal. The study provides valuable insights into the interactions between land use, landscape structure, and genetic diversity, which have significant implications for mosquito vector control and disease management strategies.
Comments and Suggestions:
The abstract provides a concise overview of the research findings. However, a few improvements can be made to enhance its clarity and impact:
- Provide More Context: While the abstract mentions genetic diversity and population structuring for Haemagogus leucocelaenus, it lacks essential context and background information. Briefly introducing the significance of yellow fever and its transmission dynamics in the Atlantic Forest would help readers better understand the relevance of the study.
- Quantify Results: Instead of stating that genetic diversity and population structuring "varied," provide specific quantitative results to highlight the extent of this variation. For example, mention the range of genetic diversity observed across different sites and the magnitude of population structuring indicated by the distance-based analysis.
- Clarify Agricultural Land Use Impact: The abstract mentions an association between genetic diversity and agricultural land use, but it is not entirely clear how this association is manifested. Provide a more explicit explanation of the relationship between agricultural land use and the observed genetic patterns.
- Explain Neutrality Tests: When mentioning the Tajima D and Fu FS neutrality tests, briefly explain what these tests are and how their results suggest recent population expansion. This will help readers without a background in genetics to understand the significance of these findings.
- Emphasize Novelty: Highlight any novel findings or contributions that the study brings to the field of research. Mentioning unique insights gained from this specific study will make the abstract more engaging and attractive to readers.
- Provide Location Context: The abstract mentions the northwest of the state, but it would be helpful to mention specific location names or regions, providing better geographic context for readers.
Introduction
- Citations/References: While the manuscript provides several citations, it would be helpful to include more recent and up-to-date references to strengthen the relevance and reliability of the information presented. Please reduce the number of citation those are not very relevant.
- Clarity on the Research Objective: The study's objective is briefly mentioned towards the end of the introduction. It would be beneficial to rephrase and emphasize the research question more explicitly at the beginning of the introduction to provide a clear sense of direction for the readers.
- Explain Sylvatic Yellow Fever: The manuscript mentions "sylvatic yellow fever" without providing a clear definition. It would be helpful to briefly explain the term for readers who might not be familiar with it.
Consider organizing the introduction in a more structured manner. For instance, you can start with a general overview of tropical forests, their biodiversity, and their importance in providing ecosystem services. Then, transition into the specific issues related to deforestation, the increase in zoonotic diseases, and yellow fever. Finally, conclude with a clear statement of the research objectives.
The introduction ends abruptly after introducing the study's objectives. It would be better to conclude the introduction with a clear transition sentence that sets up the structure of the following sections or highlights the significance of the research.
Materials and Methods
there are a few points that can be clarified and improved to enhance the understanding of the methods used:
Clarify Genetic Diversity Indices: When discussing the estimation of nucleotide diversity (θ), explicitly mention the genetic diversity indices S and πÌ‚ and explain their significance for readers unfamiliar with these terms.
Include Details on Statistical Analyses: While the statistical analyses are mentioned, some of the specific methods used are not explained in detail. For instance, provide a brief explanation of the statistical models or methods used for AMOVA, Tajima's D test, Fu's FS test, and FST calculations.
Define Buffer Radii Clearly: When defining the buffer radii (2,850 m and 5,700 m), explain why these specific values were chosen and how they relate to the expected dispersal radius of the species.
Explain Land Use Classes and Landscape Metrics: Provide a concise explanation of each land use class used in the study, and briefly describe the landscape metrics derived from the georeferenced data.
Reorganize Paragraphs: Some paragraphs contain multiple concepts and methods. Consider breaking them into smaller, more focused paragraphs to improve readability and facilitate comprehension.
Consistent Citation Style: Ensure that all references are cited consistently throughout the section according to the preferred citation style.
Grammar and Language: Check for any minor grammatical errors and ensure the language is clear and concise.
Results
there are some areas where additional clarity and organization can further enhance the presentation:
Define Genetic Diversity Indices: At the beginning of the "Results" section, briefly explain the genetic diversity indices used, such as θS and θ?Ì‚, to provide readers with a better understanding of their significance.
Provide Interpretation of Results: Offer interpretations or implications of the findings at each stage. For example, after presenting the genetic diversity indices for different populations, discuss the potential reasons for variations and what these results might mean in the context of the study.
Clarify Geographic Distances: In the section discussing the geographic distances between collection sites, specify the units used (e.g., kilometers) to provide a clear understanding of the scale.
Consider Splitting the Section: The "Genetic Structure" subsection contains information related to FST values, geographic distance, multidimensional scaling, and hierarchical cluster analysis. Consider splitting this subsection into smaller, more focused sections to improve readability and organization.
Rephrase Sentences for Clarity: Some sentences may benefit from rephrasing to improve clarity and readability.
Consistent Citation Style: Ensure that all references cited in the "Results" section adhere to the preferred citation style.
Discussion
In Discussion some areas where additional clarity and elaboration can improve the understanding of the results:
Provide a Clear Hypothesis: At the beginning of the "Discussion," explicitly state the main hypothesis tested in the study. This will help readers understand the context and purpose of the research.
Interpret Genetic Diversity Results: Elaborate on the ecological implications of the observed genetic diversity patterns in the different populations of Hg. leucocelaenus. Discuss how the variations in genetic diversity might be linked to different environmental factors, including land use and landscape structure.
Explain the Significance of Landscape Connectivity: Provide a more in-depth discussion of the role of landscape connectivity and "ecological corridors" in facilitating the dispersal of Hg. leucocelaenus. Explain how this connectivity might influence gene flow and population structuring.
Discuss Population Expansion: Further explore the reasons for population expansion observed in certain populations, particularly in São Paulo and Mairiporã. Consider potential factors contributing to this expansion, such as urbanization, land-use changes, and ecological adaptability of the species.
Address Potential Limitations: In discussing the potential limitation of the study regarding immature specimens collected in Miracatu, consider how this limitation might have affected the interpretation of results. Offer insights into how additional data from this area could further contribute to the overall findings.
Comparison with Other Studies: Compare and contrast the results of this study with relevant previous studies on other mosquito species in the same region or using similar genetic markers. This comparison can help contextualize the findings and identify potential similarities or differences in population genetics patterns.
Overall Implications and Future Directions: Conclude the "Discussion" section with a concise summary of the main implications of the study's findings. Additionally, suggest potential future research directions that could build upon the current findings and address any remaining questions or limitations.

Author Response
Dear Reviewer
Thank you very much for reviewing our manuscript and for the suggestions, which have enabled us to greatly improve the text.
We have made the changes to the manuscript to address the points, and we hope the article is now suitable for publication in Insects.
Sincerely,
Mauro Marrelli
REVIEWER 2
Authors investigated the genetic diversity and population structuring of Hg. leucocelaenus mosquitoes in various municipalities in the state of São Paulo. Using mitochondrial SNPs as molecular markers, the study revealed that genetic diversity was associated with agricultural land use, with higher diversity in regions with increased agricultural activity. Population structuring was observed, with populations in the northwest region forming a distinct clade. The results suggest landscape connectivity as a key factor in mosquito dispersal. The study provides valuable insights into the interactions between land use, landscape structure, and genetic diversity, which have significant implications for mosquito vector control and disease management strategies.
- We thank the reviewer for their time and comments which have greatly strengthened the manuscript. Below is an enumerated response to each of the comments.
Comments and Suggestions:
The abstract provides a concise overview of the research findings. However, a few improvements can be made to enhance its clarity and impact:
Response: Thank you very much for your comments and suggestions. Unfortunately, the abstract may not have been rewritten to meet the topics listed below. According to the rules of the journal, the abstract cannot be longer than 200 words, and must be written concisely. Thus, with the space restricted, we could not meet all the suggestions in this section. Suggestions have been included in the sections referred to within the main text of the article.
Provide More Context: While the abstract mentions genetic diversity and population structuring for Haemagogus leucocelaenus, it lacks essential context and background information. Briefly introducing the significance of yellow fever and its transmission dynamics in the Atlantic Forest would help readers better understand the relevance of the study.
Response: As we commented, we could not add more text in the abstract. However, information has been included in the introduction of the article, as per your suggestions below
- Quantify Results: Instead of stating that genetic diversity and population structuring "varied," provide specific quantitative results to highlight the extent of this variation. For example, mention the range of genetic diversity observed across different sites and the magnitude of population structuring indicated by the distance-based analysis.
Response: We have included more information in the results section
- Clarify Agricultural Land Use Impact: The abstract mentions an association between genetic diversity and agricultural land use, but it is not entirely clear how this association is manifested. Provide a more explicit explanation of the relationship between agricultural land use and the observed genetic patterns.
Response: We clarified this issue in the main text
- Explain Neutrality Tests: When mentioning the Tajima D and Fu FS neutrality tests, briefly explain what these tests are and how their results suggest recent population expansion. This will help readers without a background in genetics to understand the significance of these findings.
Response: We explain what these methods mean in the material and methods section.
- Emphasize Novelty: Highlight any novel findings or contributions that the study brings to the field of research. Mentioning unique insights gained from this specific study will make the abstract more engaging and attractive to readers.
Response: As we commented, we could not add more text in the abstract.
- Provide Location Context: The abstract mentions the northwest of the state, but it would be helpful to mention specific location names or regions, providing better geographic context for readers.
Response: As we commented, we could not add more text in the abstract. This information is included in the material and methods section
Introduction
- Citations/References: While the manuscript provides several citations, it would be helpful to include more recent and up-to-date references to strengthen the relevance and reliability of the information presented. Please reduce the number of citation those are not very relevant.
Response: We thank you for the comment. However, we cannot meet this request, because the citations used in this study are characterized as essential for its construction.
- Clarity on the Research Objective: The study's objective is briefly mentioned towards the end of the introduction. It would be beneficial to rephrase and emphasize the research question more explicitly at the beginning of the introduction to provide a clear sense of direction for the readers.
- Explain Sylvatic Yellow Fever: The manuscript mentions "sylvatic yellow fever" without providing a clear definition. It would be helpful to briefly explain the term for readers who might not be familiar with it.
Consider organizing the introduction in a more structured manner. For instance, you can start with a general overview of tropical forests, their biodiversity, and their importance in providing ecosystem services. Then, transition into the specific issues related to deforestation, the increase in zoonotic diseases, and yellow fever. Finally, conclude with a clear statement of the research objectives.
The introduction ends abruptly after introducing the study's objectives. It would be better to conclude the introduction with a clear transition sentence that sets up the structure of the following sections or highlights the significance of the research.
Response: Thanks for the suggestions for improving our introduction. A definition referring to the term sylvatic yellow fever has been added to the text (lines 382 – 386).
“Sylvatic yellow fever cycle is an excellent model to study, since it involves several species of non-human primates (NHPs) and mosquitoes belonging to the genera Haemagogus and Sabethes.”
While a paragraph (line 446 – 452) was added before the objectives of the work, in which we believe it fixes the closure problem in an abruptly identified way.
“As mentioned earlier, recent studies carried out in areas of the Atlantic Forest have shed light on the influence of forest fragmentation on the dispersion and circulation of YFV [32,33]. Therefore, it is essential to verify the role of forest fragmentation also in the populations of the vectors, especially in the species Hg. leucocelaenus, in view of its ecological plasticity to impacted environments.”
Materials and Methods
there are a few points that can be clarified and improved to enhance the understanding of the methods used:
Clarify Genetic Diversity Indices: When discussing the estimation of nucleotide diversity (θ), explicitly mention the genetic diversity indices S and πÌ‚ and explain their significance for readers unfamiliar with these terms.
Response. Thank you for pointing out the need for further clarification in this sentence. We have added more information as requested (lines 538 – 544).
“For nucleotide diversity, the values of theta (θ): S and , indexes which represent the variations within and between the populations, were calculated. The former is estimated from the observed number of polymorphic loci, while the latter is estimated from the mean number of pairwise differences between DNA sequences [49,50], both used to measure the degree of polymorphism present in DNA.”
Include Details on Statistical Analyses: While the statistical analyses are mentioned, some of the specific methods used are not explained in detail. For instance, provide a brief explanation of the statistical models or methods used for AMOVA, Tajima's D test, Fu's FS test, and FST calculations.
Response: Thank you for pointing out the need for further explanation. We have added more information as requested (lines 535 – 535 and 547 -549).
“ The values for nucleotide diversity within the populations and the analysis of molecular variance (AMOVA) (used to verify the genetic variation within and between groups of individuals, based on the similarity obtained by calculating the sum of squares)”
“Signs of demographic events in the populations (expansion or retraction) were identified with Tajima’s D test for neutrality [50] and Fu’s FS test [51], indexes that provide a history of the demographic events of the populations (negative values show recent events of population expansion, while positive values indicate possible bottlenecks)”
Define Buffer Radii Clearly: When defining the buffer radii (2,850 m and 5,700 m), explain why these specific values were chosen and how they relate to the expected dispersal radius of the species.
Response: We appreciate your suggestion and included more information as requested. Both rays were based on the maximum dispersion (5.7 kilometers / 5,700 meters) for the species Haemagogus leucocelaenus (DOI: 10.4269/ajtmh.1950.s1-30.301). The smaller buffers (2,850 m) corresponded to the expectation of dispersal in the landscape expected by Jackson and Fahrig (https://doi.org/10.1007/s10980-012-9757-9), in which, the dispersion of a given species will be 0.3 – 0.5 (its maximum dispersion); while the second ray was based on the maximum dispersion of the species.
“Two buffers with radii of 2,850 m and 5,700 m respectively were defined around each collection point to calculate the landscape metrics. The buffer radius was based on an estimate of the expected dispersal radius of 0.3 to 0.5 times the maximum dispersal distance of a species [61]. In the case of Hg. leucocelaenus this corresponds to 5.7 km [62].”
Explain Land Use Classes and Landscape Metrics: Provide a concise explanation of each land use class used in the study, and briefly describe the landscape metrics derived from the georeferenced data.
Response: We appreciate your suggestion and included more information as requested (lines 593 – 598)
“Subsequently, the land use classes were used to quantify the landscape metrics: Total area (CA) and Percentage of the landscape occupied (PLAND), and the Total Edge (TE) and Density Edge (DE) metrics were also calculated for the Forest Formation class.”
Reorganize Paragraphs: Some paragraphs contain multiple concepts and methods. Consider breaking them into smaller, more focused paragraphs to improve readability and facilitate comprehension.
Response: We appreciate your suggestion and divided the section in smaller paragraphs
Consistent Citation Style: Ensure that all references are cited consistently throughout the section according to the preferred citation style.
Response: We appreciate your suggestion and checked all references
Grammar and Language: Check for any minor grammatical errors and ensure the language is clear and concise.
Response: We appreciate your suggestion and checked all text for any minor grammatical errors and ensure the language is clear and concise.
Results
there are some areas where additional clarity and organization can further enhance the presentation:
Define Genetic Diversity Indices: At the beginning of the "Results" section, briefly explain the genetic diversity indices used, such as θS and θ?Ì‚, to provide readers with a better understanding of their significance.
Response: We appreciate your suggestion. However, we believe that the inclusion of such a sentence would make the text redundant, since the definition is already in the previous section "2.4. Population Diversity and Stratification Analysis".
Provide Interpretation of Results: Offer interpretations or implications of the findings at each stage. For example, after presenting the genetic diversity indices for different populations, discuss the potential reasons for variations and what these results might mean in the context of the study.
Response: We appreciate your suggestion. We discussed that is in the discussion section.
Clarify Geographic Distances: In the section discussing the geographic distances between collection sites, specify the units used (e.g., kilometers) to provide a clear understanding of the scale.
Response: We appreciate your comment. In the text the unit used is specified in kilometers (km).
Consider Splitting the Section: The "Genetic Structure" subsection contains information related to FST values, geographic distance, multidimensional scaling, and hierarchical cluster analysis. Consider splitting this subsection into smaller, more focused sections to improve readability and organization.
Response: We appreciate your suggestion. However, all the items listed within the item "Genetic Structure" are interconnected to each other, for example FST values are used together with geographical distances for comparison purposes, while clusters are responsible for visually informing the genetic structuring pattern of our data.
Rephrase Sentences for Clarity: Some sentences may benefit from rephrasing to improve clarity and readability.
Response: We appreciate your suggestion and rephrased some sentences
Consistent Citation Style: Ensure that all references cited in the "Results" section adhere to the preferred citation style.
Response: We appreciate your suggestion and checked all references
The above suggestions have been taken into consideration for the improvement of the text and improvement of the reading.
Discussion
In Discussion some areas where additional clarity and elaboration can improve the understanding of the results:
Provide a Clear Hypothesis: At the beginning of the "Discussion," explicitly state the main hypothesis tested in the study. This will help readers understand the context and purpose of the research.
Response:
The Discussion begins by stating the hypothesis of the study, as suggested (lines 744 - 747)
Interpret Genetic Diversity Results: Elaborate on the ecological implications of the observed genetic diversity patterns in the different populations of Hg. leucocelaenus. Discuss how the variations in genetic diversity might be linked to different environmental factors, including land use and landscape structure.
Response: Thanks for the suggestion. Although these points can be extensively discussed, we believe that the present text already provides some explanation of ecological implications and how genetic variations can be linked to different environmental factors. The parts mentioned are on lines 762 – 771.
Explain the Significance of Landscape Connectivity: Provide a more in-depth discussion of the role of landscape connectivity and "ecological corridors" in facilitating the dispersal of Hg. leucocelaenus. Explain how this connectivity might influence gene flow and population structuring.
Discuss Population Expansion: Further explore the reasons for population expansion observed in certain populations, particularly in São Paulo and Mairiporã. Consider potential factors contributing to this expansion, such as urbanization, land-use changes, and ecological adaptability of the species.
Address Potential Limitations: In discussing the potential limitation of the study regarding immature specimens collected in Miracatu, consider how this limitation might have affected the interpretation of results. Offer insights into how additional data from this area could further contribute to the overall findings.
Comparison with Other Studies: Compare and contrast the results of this study with relevant previous studies on other mosquito species in the same region or using similar genetic markers. This comparison can help contextualize the findings and identify potential similarities or differences in population genetics patterns.
Overall Implications and Future Directions: Conclude the "Discussion" section with a concise summary of the main implications of the study's findings. Additionally, suggest potential future research directions that could build upon the current findings and address any remaining questions or limitations.
Response: We appreciate your suggestion. We believe that all the other points raised in the discussion item are relevant and can certainly bring more information to the discussion. However, because of the small time available for the modifications and because we think that this can greatly increase the size of the text, we have chosen to make the discussion more succinct and bring references that can help those who aim to better understand some of the concepts and processes addressed in the study.
